# Multi-Organ Crosstalk with Endocrine Pancreas: A Focus on How Gut Microbiota Shapes Pancreatic Beta-Cells

**DOI:** 10.3390/biom12010104

**Published:** 2022-01-08

**Authors:** Elisa Fernández-Millán, Carlos Guillén

**Affiliations:** 1Department of Biochemistry and Molecular Biology, Faculty of Pharmacy, Complutense University of Madrid, 28040 Madrid, Spain; elfernan@ucm.es; 2Spanish Biomedical Research Centre in Diabetes and Associated Metabolic Disorders (CIBERDEM), Instituto de Salud Carlos III, 28040 Madrid, Spain

**Keywords:** pancreatic beta-cells, type 2 diabetes, inter-organ communication, intestine, short-chain fatty acids, extracellular vesicles

## Abstract

Type 2 diabetes (T2D) results from impaired beta-cell function and insufficient beta-cell mass compensation in the setting of insulin resistance. Current therapeutic strategies focus their efforts on promoting the maintenance of functional beta-cell mass to ensure appropriate glycemic control. Thus, understanding how beta-cells communicate with metabolic and non-metabolic tissues provides a novel area for investigation and implicates the importance of inter-organ communication in the pathology of metabolic diseases such as T2D. In this review, we provide an overview of secreted factors from diverse organs and tissues that have been shown to impact beta-cell biology. Specifically, we discuss experimental and clinical evidence in support for a role of gut to beta-cell crosstalk, paying particular attention to bacteria-derived factors including short-chain fatty acids, lipopolysaccharide, and factors contained within extracellular vesicles that influence the function and/or the survival of beta cells under normal or diabetogenic conditions.

## 1. Introduction

The prevalence of type 2 diabetes (T2D) is rapidly accelerating all over the world and the low- and middle-income countries achieve more than 75% of people with diabetes [1]. This rising incidence is a consequence of longer life expectancy, unhealthy diets, and epigenetic modifications. Clearly, poor dietary habits play a predominant role in the appearance of obesity, low-grade chronic inflammation, and insulin resistance which drive slow progression from prediabetes to overt T2D. In this respect, pancreatic beta-cell dysfunction is one of the hallmarks of T2D. Once insulin resistance appears, beta cells compensate for the increased insulin demand by enhancing their proliferation, mass, and/or function in order to maintain normoglycemia. In fact, body mass index positively correlates with beta-cell mass [2,3]. However, as the disease progresses, the chronically increased workload on beta cells results in cell exhaustion and the remaining beta cells are no longer capable of correctly secreting insulin [4]. Recent research has pointed out that beta-cell dysfunction is an early event preceding the reduction in beta-cell mass in prediabetic patients [5]. Accordingly, most autopsy studies in pancreas from T2D donors have revealed that beta-cell mass is decreased up to 50% in comparison to healthy individuals [3,6]. This reduction is mostly due to increased inflammation and apoptosis rather than a reduction in the rate of beta-cell proliferation [6,7]. The mechanisms underlying the loss of functional beta-cell mass have been extensively studied in the last decades. Classically, the term glucolipotoxicity was coined to explain how, in the diabetic milieu, chronic excess of nutrients such as glucose, free fatty acids (FFA), and other lipid intermediates synergistically induce deleterious effects on both beta-cell mass and function [8,9]. However, it is worth noting that the maintenance of systemic homeostasis and the capacity to face nutritional and environmental challenges is a global process and require the coordination of multiple organs and tissues. In recent years, it has increasingly been recognized that circulating levels of various factors from other organs, independent of glucose, or obesity, may also influence beta-cell function and plasticity. These secreted factors include hormones, cytokines, growth factors, or small molecules that relay important information about metabolic flux between physically distant cell types and the endocrine pancreas. Advances in proteomics and lipidomics have allowed identifying how the profile of these secreted factors shift during insulin-resistant conditions. Moreover, improved sequencing and advanced-data processing have also provided insights into coding and non-coding RNAs (such as miRNAs) with the potential to affect beta-cell function and/or survival. In this context, the study of extracellular vesicles (EVs) is an active area of research. EVs are membrane-bound extracellular compartments containing bioactive molecules which secretion from cells is a controlled process known to respond to nutrient-related signals thus suggesting that this method of communication may play a role in response to changing metabolic states [10,11]. In support of this idea, enough evidence points out that EVs play a role in obesity and the regulation of peripheral insulin sensitivity, a major component of the pathogenesis of T2D [12,13,14], by serving as a mode of intercellular communication among adipose tissue, liver, or skeletal muscle. In agreement, higher levels of circulating EVs are described in both animal models and patients of T2D compared to healthy individuals [11,15]. However, EVs and their bioactive cargo can also significantly affect the capacity of pancreatic beta cells to produce and secrete insulin, or they may also impact beta-cell survival through modulation of proliferative, inflammatory, and apoptotic pathways [16]. In this complex multi-organ crosstalk, abundant literature highlights the role of gut microbiota-derived metabolites and beta-cell function. Although, the EV-mediated crosstalk between gut and pancreatic beta-cells remains less explored its study may also yield unexpected insights that are now only beginning to be appreciated.

Overall, the findings summarized in this review highlight the importance of inter-organ communication with pancreatic beta-cells in both, health and disease. They further illustrate that there are multiple stimuli, including the type of diet, modifying the expression, secretion, or function of tissue-derived factors with the potential to affect beta-cell function and/or survival and therefore promoting glucose homeostasis deregulation.

## 2. Inter-Organ Crosstalk Impacting Beta-Cell Function and Mass

The effects of insulin secretion on metabolic organs such as the liver, skeletal muscle, and adipose tissue are clearly relevant for carbohydrate metabolism by improving glucose uptake or utilization as well as by inhibiting hepatic glucose output and hence, reducing hyperglycemia. However, this is a bidirectional event since individual beta cells can sense a multitude of signals derived from other tissues that are integrated into physiological beta-cell responses to metabolic demand but that can also adversely affect beta cells by impairing their functions and survival (Figure 1).

### 2.1. Fat-Cell to Beta-Cell Communication

Beyond its function as an energy reservoir, the adipose tissue must be considered a complex and dynamic endocrine organ which in healthy conditions have a positive systemic effect through the release of lipids, metabolites, adipokines, or extracellular vesicles. However, obese individuals show dysregulated secretion of adipocyte-derived factors contributing to low-grade chronic inflammation and insulin resistance [17,18]. One of the first adipokines to be associated with direct beta-cell effects was leptin. The hormone leptin was primarily recognized for its actions in the central nervous system to regulate food intake and energy expenditure. However, beta cells also express the leptin receptor, LepR (Ob-R), in both the functional long form (LepRb) and truncated shorter forms [19]. Leptin suppresses insulin secretion by promoting the surface abundance of KATP channels, which causes membrane hyperpolarization and renders beta cells electrically silent [20,21]. Moreover, this hormone has the additional effect of reducing pre-proinsulin gene expression, both in vitro and in vivo [22].

Adipsin, an adipokine that controls the generation of the complement component C3a, is shown to be decreased in patients with T2D. However, recent work has demonstrated that replenishment of its levels improves hyperglycemia and increases insulin levels in diabetic mice without long-term beta-cell failure associated with other insulin secretagogues [23]. One of the main adipocyte-derived hormones, adiponectin, is able to act as a trophic factor inducing beta-cell proliferation under conditions of pharmacological beta-cell destruction [24].

Another important mediator of intercellular communication between adipose tissue and beta cells is extracellular vesicles (EVs). Gesmundo et al. [13] recently observed that EVs derived from healthy adipocytes, obtained in vitro from mouse 3T3-L1 cells or ex vivo from human adipose tissue, when applied to rat INS-1E/human EndoC-βH3 beta cells or human islets, contributed to normal beta-cell physiology and protected them from palmitate or pro-inflammatory cytokines induced injury and cell death. On the contrary, exposure of beta cells to EVs from obese or inflamed adipocytes resulted in impairment of insulin secretion and cell death. These opposite effects on beta cells seemed to be partly associated with differential expression of miRNAs between EVs from healthy or inflamed adipocytes. miRNAs are non-coding RNAs that function to induce the degradation or inhibition of translation when acting on their target mRNAs. Accordingly, in EVs from inflamed adipocytes the authors described upregulation of miR-155, miR-30, and miR-146, known to be involved with insulin secretion, proliferation, or apoptotic pathways.

### 2.2. Hepatocyte to Beta-Cell Communication

Being an essential organ in the storage of energy as well as in glucose and lipid metabolism, the liver has a central role in the development of insulin resistance and T2D. Interestingly, the liver secretes proliferative factors promoting compensatory hyperplasia of islet-beta cells during obesity and insulin resistance [25,26]. In agreement, inducible liver insulin receptor knockout mice (iLIRKO), a model of severe hepatic insulin resistance, shows significant beta-cell mass enhancement in parallel to elevated levels of hepatic and circulating insulin-like growth factor-1 (IGF-1), a known trophic factor for beta cells [26]. In fact, IGF-1 levels were correlated to the level of hepatic insulin resistance [26].

In advanced T2D patients, defective beta-cell function leads to a lack of paracrine regulatory factors that normally modulate glucagon release (i.e., insulin) and the resulting glucagon hypersecretion worsens the elevation of liver glucose output associated with the lost inhibitory effect of insulin, even in the fed state. In this context, glucagon signaling via PKA induces kisspeptin1 production in hepatocytes which impacts beta-cell function through the kisspeptin1 receptor (Kiss1R, also known as GPR54). Conclusions regarding the effect of kisspeptin1 on beta cells are controversial since increased kisspeptin1 levels in the liver and plasma of both obese mouse models and T2D patients have been described to inhibit insulin secretion [27] whereas placenta-derived kisspeptin1 seems to play a physiological role in the islet adaptation to pregnancy, maintaining maternal glucose homeostasis by acting through the beta-cell GPR54 receptor. Indeed, circulating kisspeptin1 levels were significantly lower in women with gestational diabetes [28]. Similarly, EVs released by hepatocytes appear to determine islet-beta cell expansion capacity. Under obesogenic diet, miRNA cargo profile of hepatocyte-derived EVs changed, with diminished miR-7218-5p levels, which resulted in increased expression of pro-proliferative CD74 in the beta-cell line MIN6 [29]. This way of communication between the liver and pancreas may be a potential mechanism for islet beta-cell compensatory hyperplasia in obesity and insulin resistance.

### 2.3. Muscle-Cell to Beta-Cell Communication

Physical activity and/or exercise are essential to the prevention and treatment of T2D [30,31]. Noteworthy, the health benefits of exercise are more than increased energy expenditure or improved energy balance since skeletal muscle is also an endocrine organ that produces and secretes biological mediators (myokines) able to regulate the function of other tissues and organs such as the pancreas. IL-6 is one of the best-characterized myokines produced in an exercise setting which cross-talks to beta cells potentiating glucose-stimulated insulin secretion. Ellinsgaard et al. [32] demonstrated in mice that this effect on beta cells was not direct but mediated by promotion of GLP-1 production and release from alpha cells through increased expression of proglucagon and prohormone convertase 1/3 to favor GLP-1 posttranslational processing rather than glucagon. It has yet to be confirmed whether IL-6 stimulates insulin secretion in humans.

It should be noted, however, that IL-6 is a cytokine with many activities including the regulation of the immune system and activation of pro-inflammatory signals [33]. Then, in contrast to the beneficial effects of exercise-induced acute raise of systemic IL6, chronically elevated IL-6 levels in patients with obesity or T2D [34] might cause detrimental effects on beta-cell survival similarly to elevated IL-6 in the liver contributes to hepatic cell dysfunction and insulin resistance [35]. In addition to IL-6, conditioned media from insulin-resistant human myotubes also contained elevated levels of other myokines such as CXCL10 that, when added to primary human and rat islets, significantly decreased proliferation and increased apoptosis of beta cells [36].

Other molecules with a potential role in skeletal muscle-to-beta-cell crosstalk effects are miRNAs, which can be released from muscle in circulation either directly, or into muscle-derived EVs (reviewed in detail by Barlow and Solomon) [37]. During exercise or in T2D there is a shift in muscle-specific miRNA profile [38,39]. Jalabert et al. [40] demonstrated that isolated mice islets or MIN6B1 cells were able to take up skeletal muscle-derived exosomes from insulin-resistant mice fed a palmitate-enriched diet. This specific cargo affected beta-cell gene expression signature and increased their proliferation suggesting a novel mechanism whereby miR-16 regulates the adaptive response of beta cells during the development of obesity-induced T2D.

### 2.4. Gut to Beta-Cell Communication

In the last decades, numerous studies have evidenced the capacity of beta cells to interact with diverse non-metabolic tissues as the gut. This is providing new insights into possible strategies for improving beta-cell function and/or mass which undoubtedly might have beneficial effects for patients with diabetes.

In the early 1900s emerged the concept that certain factors produced by the gastrointestinal tract after nutrient ingestion were able to stimulate the release of insulin from the endocrine pancreas and thereby, reduce blood glucose levels [41]. These glucose-lowering intestinal factors were called “incretins” [42] and account for approximately 50–70% of the total insulin secreted after oral glucose administration, a phenomenon known as the “incretin effect” [43]. The first incretin hormone to be identified was the glucose-dependent insulinotropic polypeptide (GIP) which is produced and released from intestinal K-cells in response to glucose and lipids. However, it soon became evident that the incretin effect was not only due to GIP activity, but other incretin hormone released from ileum L-cells, glucagon-like peptide-1 (GLP-1) contributed to postprandial insulin secretion [44]. Beyond their effects on the functionality of beta cells, both peptides are capable of inducing beta-cell proliferation and resistance to apoptosis, therefore, increasing beta-cell mass. GLP-1 activates the expression of the Pdx-1 transcription factor that regulates islet-cell proliferation and differentiation [45]. Accordingly, GLP-1R−/− mice exhibit defective regeneration of beta-cell mass and deterioration of glucose tolerance after partial pancreatectomy and are more susceptible to streptozotocin-induced beta-cell depletion [46]. The discovery that the insulinotropic properties of GLP-1 are preserved in human subjects with T2D [47] together with the capacity of GLP-1 to inhibit glucagon secretion, food intake, or gastric emptying, prompted the development of GLP-1 mimetics and inhibitors of GLP-1 degradation by dipeptidyl peptidase 4 (DPP4) for the treatment of T2D (reviewed in [48]). On the other hand, the increasing evidence of elevated postprandial GLP-1 levels observed in obese people after Roux-Y gastric bypass surgery, together with amelioration of hyperglycemia and insulin resistance, strongly suggests benefits of recruiting endogenous GLP-1 as an alternative treatment [49]. Thus, in the past two decades, many efforts have been made on elucidating the role of nutritional components, either directly or through bacterial-end products in regulating GLP-1 secretion [44,50] and thereby, beta-cell biology. This issue is discussed hereafter.

## 3. Diet Shapes Gut Microbiota

The term ‘gut microbiota’ refers to the trillions of microorganisms that colonize the intestine and provide us with genetic and metabolic properties relevant to the maintenance of our body homeostasis. However, an aberrant gut microbiota composition or dysbiosis is associated with several diseases, including obesity, diabetes, or inflammatory bowel disorder [51,52]. Factors including age, genetics, and diet may influence microbiome composition. Of these, diet is the primary modulator of bacterial richness and abundance since its macronutrient (fat, sugar, or protein) and fiber content determine the type of microbial-derived metabolites and subsequently its health outcomes.

Dietary fibers which escape digestion by host enzymes in the upper gut are metabolized by the microbiota in the cecum and colon, mainly generating short-chain fatty acids (SCFAs). The most abundant SCFAs are acetate, propionate, and butyrate. Butyrate is the primary energy source for colonocytes and is locally consumed, whereas the other SCFAs are absorbed draining into the portal vein. Propionate is metabolized in the liver as a substrate for gluconeogenesis and acetate may be used as cholesterol or fatty acid precursors [53,54]. Interestingly, besides providing 5–10% of energy to the host as metabolic substrates, SCFAs also act as signaling molecules binding and stimulating G-protein coupled receptors (GPR) 41/43 (also known as FFA3/2, respectively). In rodent and human intestinal cell lines, SCFAs trigger the secretion of peptide YY (PYY) along with glucagon-like peptide 1 (GLP-1) [44,55]. The same effect has been reported in vivo [50,55]. In this way, SCFAs can indirectly influence host appetite at the hypothalamic level but also regulate glucose-stimulated insulin secretion from pancreatic beta cells. However, unlike GPR41 knock-out mice, GPR43 knock-out animals showed markedly downregulated GLP-1 in circulation, indicating that GPR43/FFA2 plays a more important role in these effects [44,50].

SCFAs are also important for preserving epithelial barrier function. Butyrate increases mucus production and also regulates the expression of tight-junction proteins zonula occludens (ZO)-1 and occludin, contributing to reduced intestinal permeability [56,57]. However, when fermentable fibers are in short supply, such as in high-fat diets, the reduced fermentative activity of the microbiota leads to the appearance of SCFAs as minor end products. This event has been associated with disruption of tight-junction assembly, increased intestinal permeability, and lipopolysaccharide (LPS) leakage into the portal blood circulation triggering chronic subclinical inflammatory process which promotes insulin resistance through activation of toll-like receptor 4 (TLR4) in peripheral tissues [58,59]. On the contrary, selective modulation of gut microbiota by prebiotics enhanced gut barrier function during obesity and T2D being this effect partly attributed to an increase in endogenous GLP-2 production [60]. Similarly, pharmacological treatment with GLP-2 decreased gut permeability as well as systemic LPS content, which finally blunted the inflammatory state of ob/ob mice [61].

However, the beneficial effects of SCFAs on host metabolism are not only limited to their local actions in the intestine, but also related to the activation of their receptors on the liver, adipose tissue, brain, and pancreas. Thus, to better identify new therapeutic targets focused on improving beta-cell function and survival, it will be a major challenge to pinpoint the precise signaling cascades triggered by SCFAs in beta cells. Moreover, other ways of communication between the gut and endocrine pancreas might also be linked to initiation and aggravation of T2D.

## 4. Gut Microbiota Shapes Pancreatic Beta-Cells

### 4.1. Modulation of Islet Responses by Short-Chain Fatty Acids (SCFAs)

There are multiple factors that could modulate pancreatic beta-cell function. Among them, butyrate, propionate, and acetate (known as SCFAs), which derives from the microbial metabolism, are essential in controlling important metabolic processes and are involved in organ intercommunication. Among the many actions of SCFAs on pancreatic beta cells, we will focus on three of them, which are insulin secretion, proliferation, and apoptosis. Interestingly, SCFAs are able to modulate one of the key functions in pancreatic beta cells, which is insulin secretion. In fact, the detection in these cells of different receptors for SCFAs has revealed a new therapeutic window for treating T2D. In this regard, and as it has been previously described, there are mainly two types of receptors (FFA2 or GPR43 and FFA3 or GPR41) which belong to the G-protein coupled receptors (GPCRs) [62]. While GPR41 couples only to Gαi to decrease intracellular cyclic AMP levels, GPR43 is able to recruit both Gαi and Gαq to decrease cyclic AMP and increase diacylglycerol and inositol trisphosphate generation, respectively. The data published so far do not provide a consensus role for SCFAs and their receptors in beta cells. It has been observed that the deletion of both GPR43 and GPR41 is associated with increased insulin secretion and an improvement in glucose tolerance [63]. However, other authors have noticed that propionate and acetate can facilitate pancreatic beta-cell function by an increase in insulin secretion in human islets in vitro mainly through GPR43 [64,65]. Similarly, McNelis et al. [66] demonstrated that acetate is able to increase insulin secretion in wild-type beta cells, but it has an inhibitory effect in beta cells from GPR43 knock-out mice suggesting that this action could be mediated by GPR41 activation. These apparently controversial data might also respond to the possibility of forming GPR41/43 heteromers in those cells co-expressing both receptors [67]. This could be the case of pancreatic beta cells where a putative GPR41/43 heteromer might mediate the phenotypes observed in GPR41/43 single-knockout studies [63,65,66] because the knockout of either GPR41 or GPR43 would lead to the loss of the heteromer and any specific heteromer-dependent functions. In addition to these effects, SCFAs produced by the microbiome could contribute to obesity in children [68]. Then, as a consequence of obesity and insulin resistance generation, they could be involved in an overactivation of pancreatic beta cells, in order to compensate for insulin resistance [51,68]. In this regard, data obtained in humans and in animals suggests beneficial effects of acetate through the production of different hormones regulating appetite, lipolysis, and inflammation. Altogether, these data clearly indicate that a more profound investigation is needed for determining the whole effect and the interconnection between the different SCFAs in the balance of metabolic homeostasis [69]. Very interestingly, although the more abundant SCFA is acetate, different gut commensals bacteria can convert acetate into butyrate and propionate, increasing the complexity of the possible effects in general in the whole organism and, specifically in pancreatic beta cells [70]. The main effects of the different SCFAs on insulin secretion by pancreatic beta cells are shown in Figure 2.

Another two important functions of SCFAs, as was mentioned before, are the regulation of pancreatic beta-cell proliferation and the control of beta-cell death. After the treatment of mice with a high-fat diet (HFD), an increase in both GRP43 and acetate levels is observed in the islets. However, in the GRP43 knock-out animals fed with HFD, an insulin secretion defect with a concomitant glucose intolerance is observed. Similarly, after the stimulation of pancreatic beta cells MIN6 with a GPR43 agonist, insulin secretion is enhanced. All these data suggest a positive net effect of the GPR43 receptor in the control of insulin secretion both in vivo as well as in vitro [66]. GPR43 is particularly stimulated after acetate exposure and may act as a modulator of pancreatic beta-cell function, playing a role in the regulation of pancreatic beta-cell mass. In this regard, the elimination of the GPR43 receptor provokes a decreased pancreatic beta-cell mass at birth and during adulthood, suggesting a key role of this receptor in the control of beta-cell mass [71].

Furthermore, acetate is involved in the posttranslational modification (PTM) of different proteins, connecting metabolism with transcription and cell growth [72]. In addition, acetate is converted into acetyl-CoA through the action of acetyl-CoA synthetases (ACSs) or acetate-CoA ligases, regulating global histone acetylation, altering DNA replication and transcriptional activity [73]. Increased acetyl-CoA levels are associated with mTORC1 stimulation in response to different stimuli such as leucine levels [74]. In pancreatic beta cells, an increased acetylation status of tuberous sclerosis complex 2 (TSC2), a negative regulator of mTORC1, is involved in the stimulation of mTORC1, controlling pancreatic cell proliferation [75]. In addition, in a streptozotocin-induced diabetic model, both acetate and butyrate have been involved in an enhancement of pancreatic beta-cell proliferation and a reduction in apoptosis with other effects in several parameters, including diminished oxidative stress with a concomitant decrease in mitochondrial dysfunction [76].

Then, the microbiota is able to generate changes, through SCFAs, in histones and in the epigenetic machinery of the host, such as the inhibition of histone deacetyl-transferases (HDACs), in the case of butyrate and propionate, or the modification of histones through acetylation and crotonylation [77,78]. However, the effects of SCFAs on insulin secretion and pancreatic beta-cell growth are controversial and more studies are necessary for a better understanding of SCFAs and their regulation in the control of inter-organ communication [79].

There have been related alterations in microbiota with autoimmune pancreatitis in experimental models, suggesting that indeed dysbiosis is responsible, at least in part, for inflammatory processes such as pancreatitis [80]. However, not only in pancreatitis but changes in the microbiome have been associated with different diseases such as type 1 diabetes and even pancreatic cancer [81]. In addition, the modification of the production of SCFAs is involved in the progression of different diseases such as T2D [82] as well as alterations in the homeostasis and the correct communication between different tissues [83].

### 4.2. Modulation of Islet Responses by Extracellular Vesicles (EVs)

EVs represent an increasingly recognized and very important way of communication among different tissues in the organism. In this regard, this strategy is extremely efficient and is able to coordinate different organs including gut–brain [84,85], and pancreas–brain [86]. As was mentioned earlier, this communication is bidirectional, generating a higher level of complexity, playing an important role in both normal physiology as well as in disease [84].

The intercommunication through the production of EVs in the gut represents a mechanism by which different proteins, lipids, aggregates, or even miRNAs produced by intestinal epithelial cells or resident bacteria, can have local effects or even be transported to different locations, modulating the activity in the recipient cells. Then, depending on the cargo, the effects on the target cells can vary, altering their normal activity [87]. In fact, the modulation of the gut microbiota, through different mechanisms such as the diet, milk, or other factors, can affect the production of microbiota-produced EVs, regulating obesity and T2D, being a very attractive therapeutic target [12,14,88]. Then, microbiome modulation by different strategies including the use of prebiotics and probiotics can modify the progression of several metabolic diseases [89]. As it has been described before, there are a plethora of agents that affect the microbiome, regulating disease progression. Importantly, different cells from the intestine, such as enterocytes, intestinal epithelial cells, colonocytes, and many others, which are involved in the maintenance of the intestinal barrier, are able to generate exosomes [90,91]. For a better understanding of this topic, a recent review has been published analyzing the effect of different EVs produced by different tissues throughout the body involved in the control of pancreatic beta-cell function [16]. Then, any change in the microbiome, altering SCFA and EV production, could modify the secretion profile of the colonocytes (or other cell types), affecting the progression of the diseases. Figure 3 shows the main players that potentially regulate microbiota that could exert different effects in the intestinal cells, providing a link between gut microbiota and other organs, including the pancreas (Figure 3).

Interestingly and as was mentioned earlier, the microbiota can produce EVs with different compositions regulating key events in disease progression and in health maintenance [92,93]. In fact, the EVs derived from bacteria are known as microbiota-released extracellular vesicles (MEVs). These structures can be incorporated into different cells, including intestinal cells, or even cross the intestinal epithelium, facilitating accessibility to any location in the body [94].

Table 1 highlights the effect of different extracellular vesicles on pancreatic beta-cell function. In addition, it includes the origin of these extracellular vesicles and the different cargo determined in the interior of the structures.

## 5. Concluding Remarks

In this review, we have summarized the different ways of interaction between pancreatic beta cells with different tissues, including the gut, muscle, liver, and fat, for the maintenance of whole-body homeostasis. In addition, we have shown that when this interconnection is altered, it facilitates the appearance or even the progression of different disorders, such as metabolic diseases and cancer. Furthermore, we have shown the main factors that could change or modify the intestinal barrier, affecting the production of different molecules such as SCFAs and the production of exosomes that could lead to changes in pancreatic beta-cell function in two different ways, affecting pancreatic beta-cell survival and/or insulin secretion. Then, we highlighted changes in microbiome composition by different factors such as diet, milk consumption, and the use of prebiotics and probiotics and hence, the capacity of both exosome and SCFAs production. Overall, these changes could modify both the intestinal cell transcription profile, by changes in the epigenome through SCFAs and in the capacity of generating extracellular vesicles, ultimately affecting pancreatic beta cells.

## Figures and Tables

**Figure 1 biomolecules-12-00104-f001:**
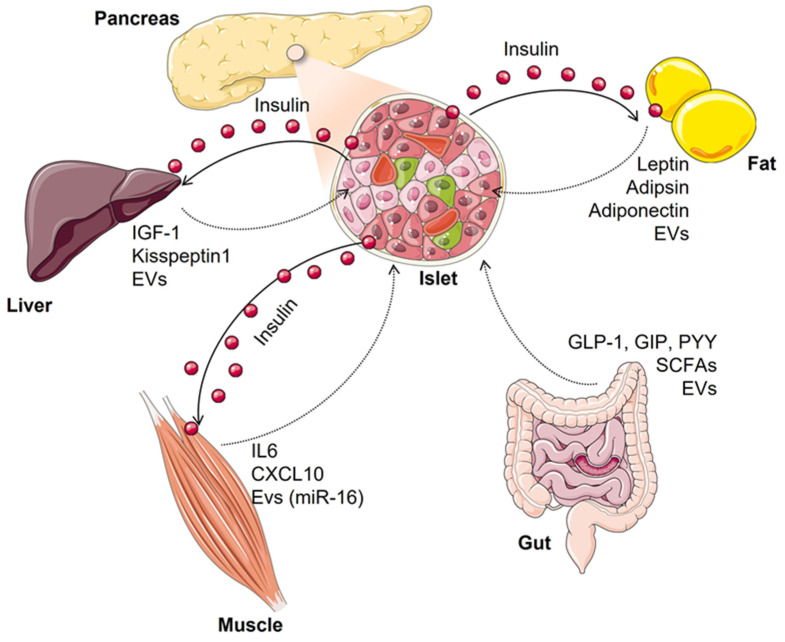
Model of crosstalk between diverse organs and pancreatic beta cells under both physiological and insulin-resistant states. The different factors represented may have the potential to enhance glucose-stimulated insulin secretion or beta-cell proliferation (as IGF-1—insulin-like growth factor-1; GLP-1—glucagon-like peptide-1; kisspeptin1 and SCFAs—short-chain fatty acids) or negatively contribute to reduce functional beta-cell mass during type 2 diabetes progression (as leptin, CXCL10 and EVs—extracellular vesicles—from inflamed adipocytes or insulin-resistant muscle cells). Each pathway is described in more detail along the text.

**Figure 2 biomolecules-12-00104-f002:**
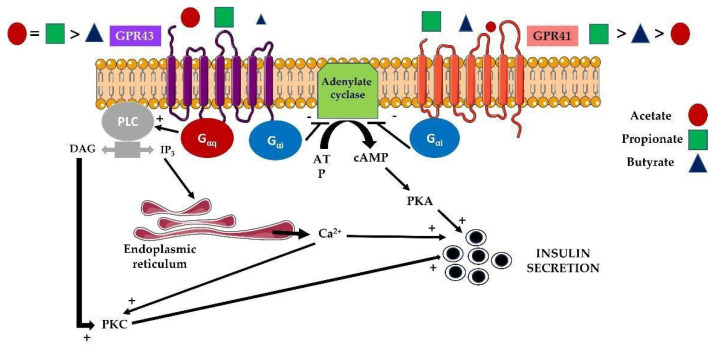
Effect of the different SCFAs on the different receptors in pancreatic beta cells regulating insulin secretion. Both GPR43 and GPR41 are GPCRs acting through two alternative pathways; Gαq and with the concomitant activation of phospholipase C (PLC) with the hydrolysis of PIP2 into diacylglycerol (DAG) and inositol triphosphate (IP3). DAG is able to activate protein kinase C (PKC) and IP3 goes to the endoplasmic reticulum to stimulate IP3 receptors and permit Ca^2+^ exit to the cytosol, where it can activate PKC. This signaling pathway activates insulin secretion. However, the alternative pathway, through Gαi is able to inactivate adenylate cyclase, decreasing the level of cAMP, not permitting the activation of protein A (PKA) which facilitates insulin secretion. Then, GPR43 has alternative signaling and GPR41 presents an inhibitory effect on insulin secretion in pancreatic beta cells. Furthermore, it is represented in the figure the differential ligand preference of SCFAs to the two different receptors.

**Figure 3 biomolecules-12-00104-f003:**
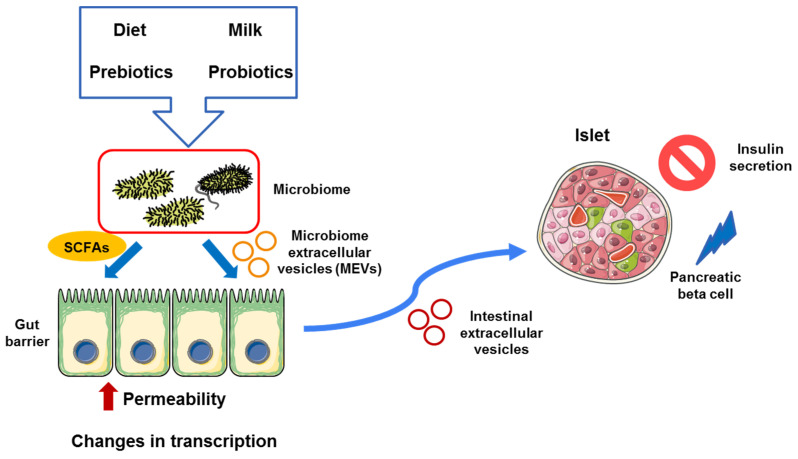
Interaction among intestinal microbiota and gut cells through the production of SCFAs and microbiome-derived exosomes during an inflammatory state, such as obesity or T2D. Then, these signals generate a response in the intestinal cells, modulating inflammation and permeability, facilitating dysbiosis, which can alter pancreatic beta-cell function by the modulation of insulin secretion and pancreatic beta-cell death/survival. In addition, several factors that can affect the microbiome including the diet, probiotics, and prebiotics, among others, modulate the capacity of production of different SCFAs and extracellular vesicles, which can potentially regulate either positively or negatively gut barrier, affecting other tissues such as pancreatic beta cells.

**Table 1 biomolecules-12-00104-t001:** Bioactive cargoes in EVs from diverse tissues and organs that can influence pancreatic beta-cell functionality and/or viability.

Tissue Origin	Model	Cargo	Effect on Pancreatic β Cells	Reference
Adipose tissue macrophages	Obesity in vitro and in vivo	miR-155	Increased in proliferation Decreased in insulin secretion	[95]
Adipocytes	Inflammation	Multiple miRNAs	Decreased cell survival Decreased insulin secretion	[13]
Mesenchymal stem cells (human umbilical cord)	T2DM	Not determined	Reversion of insulin resistance Decrease in pancreatic β cell death	[96]
Mesenchymal stem cells derived from adipose tissue	T2DM	Not determined	Increased β cell viability	[97]
Bone marrow	Hyperglycemia by STZ injection	miR106b-5p miR-222-3p	Promote pancreatic β cell proliferation	[98]
Skeletal muscle	T2DM	miR-16	Increased pancreatic β cell proliferation Increased in insulin secretion	[40]
Hepatocytes	Obesity (T2DM)	miR-7218-5p	Increased in pancreatic β cell proliferation	[29]
Not determined	STZ-treated animals+/−HFD	miR-223	Increased in pancreatic β cell proliferation Increased in insulin secretion Decreased β cell apoptosis	[99]
T-lymphocytes	T1DM	miR-142-3p miR-142-5p miR-155	Stimulates pancreatic β cell apoptosis	[100]

## Data Availability

Not applicable.

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
