# Peer review of "Multi-Organ Crosstalk with Endocrine Pancreas: A Focus on How Gut Microbiota Shapes Pancreatic Beta-Cells"

_biomolecules, 2022, doi:10.3390/biom12010104_

Round 1

Reviewer 1 Report

The manuscript focused on intercommunication between pancreatic beta-cell and some tissues (particularly the intestine) during the progression of type 2 diabetes. The topic is interesting and has a scientific contribution to the research area of type 2 diabetes. I have some comments, which can be seen below.   

  1. In general, the current review was too general, lacking detailed descriptions and deep discussion.
  2. The authors introduced the crosstalk between pancreatic beta-cell and metabolic organs, such as liver, skeletal muscle, and adipose tissue. But, the title of the manuscript only mentioned intestine-pancreatic beta-cell intercommunication, which was not reasonable. Why did authors give particular attention to the crosstalk of intestine and pancreatic beta-cell?
  3. The part of the Introduction was insufficient. Authors should emphasize the necessity and significance for investigating intercommunication between pancreatic beta-cell and some organs (particularly intestine), not simply introduce the key role of the pancreatic beta-cell. Also, please give the novelty of this review in the Introduction.
  4. Please add some tables to summarize some important findings.
  5. Please add more figures to display the key molecular mechanism.

Author Response

Manuscript ID 1531325

REFEREE 1

Comments and Suggestions for Authors

           Answers by the authors

The manuscript focused on intercommunication between pancreatic beta-cell and some tissues (particularly the intestine) during the progression of type 2 diabetes. The topic is interesting and has a scientific contribution to the research area of type 2 diabetes. I have some comments, which can be seen below.   

We would like to thank the Reviewer for the careful reading of the manuscript and for the compliments and constructive comments. All comments and suggestions have been accepted and we believe that the manuscript has been largely improved by the changes introduced.

All changes have been highlighted in red. Deleted sentences or paragraphs are indicated in red and crossed out.

  1. In general, the current review was too general, lacking detailed descriptions and deep discussion.

In order to discuss the most relevant points in more detail, some paragraphs have been rewritten in the revised manuscript (please see P4, L124-137; P7, L288-304; P10, L400-413).

    2. The authors introduced the crosstalk between pancreatic beta-cell and metabolic organs, such as liver, skeletal muscle, and adipose tissue. But, the title of the manuscript only mentioned intestine-pancreatic beta-cell intercommunication, which was not reasonable. Why did authors give particular attention to the crosstalk of intestine and pancreatic beta-cell?

We are totally agree with the Reviewer that the title chosen does not accurately reflect the content of the manuscript. For this reason (and also following the indications from Referee 2), we have changed the title to a more appropriate one:

‘Multi-organ crosstalk with endocrine pancreas: a focus on how gut microbiota shapes pancreatic beta-cells’

However, our aim has always been to place special emphasis on the role of communication between the gut and the endocrine pancreas in the setting of type 2 diabetes. That is why we have made an independent section addressing this issue and discussing the factors that allow such type of communication.

3. The part of the Introduction was insufficient. Authors should emphasize the necessity and significance for investigating intercommunication between pancreatic beta-cell and some organs (particularly intestine), not simply introduce the key role of the pancreatic beta-cell. Also, please give the novelty of this review in the Introduction.

Following Referee’s suggestions, the introduction has been more extensively explained and a sentence indicating the novelty of this review has also been added (please see P2, L49-82). 

    4. Please add some tables to summarize some important findings.

As suggested by the Reviewer, a table (Table 1) summarizing extracellular vesicle-derived cargo affecting beta-cell function and/or viability has been added in the revised version of the manuscript to better illustrate the important findings discussed along the text.

  5. Please add more figures to display the key molecular mechanism.

A new figure (Figure 2) has been included in the revised version of the manuscript depicting last findings found in literature regarding the mechanism of action of SCFAs in pancreatic beta cells. Please, note that previous Figure 2 is now renumbered as Figure 3.

Reviewer 2 Report

The review “Intestine-pancreatic beta cell intercommunication: a new approach to understand diabetes” presents an overview of the crosstalk between beta-cells and metabolic and non-metabolic tissues, and gives an update of the state of the art about microbiota role in beta-cell biology.

The review is comprehensive and clearly written, although the modifications suggested below could significantly improve the manuscript.

The title : it is focused on Intestine-pancreatic intercommunication, but the first 4,5 pages are about adipose tissue, liver and muscle dialogue with beta-cells. This is not a criticism because this part of the review is relevant and interesting, however it questions the relevance of the title. I would suggest something more general such as : “ Multi-organ crosstalk with endocrine pancreas : a focus on how gut microbiota shapes pancreatic beta-cells”, for example.

Then, the first appearance of the EVs is in Figure 1 legend (abstract excluded), which is inappropriate.

A specific paragraph on Extracellular Vesicles should take place in the Introduction, to define what it is…It seems more relevant to explain it first as it concerns the 4 tissues of interest (fat, muscle liver and gut) than to add information about EVs in page 7, in a specific Gut-centred paragraph. However, the information about microbiota-produced EVs is relevant in the 4.2 paragraph, of course.

In addition, the fact that some EVs are microbiota-produced is very important and may deserve a specific paragraph with appropriate references.

Figure 2 : the figure itself and the legend present only negative effects of SCFAs and EVs, which doesn’t correspond to the text. It is unclear if the figure displays a pathologic condition such as obesity…And the blue arrow with ‘prebiotics and probiotics” is confusing, although I understand it is a way to counterbalance the deleterious effects of dysbiosis…

Besides these general comments about the manuscript organisation, I have some specific points to discuss.

P2, L52-53 : I feel that the sentence doesn’t take into account the specific effect of insulin on the liver, that is to say the decreased hepatic glucose production. Both improving glucose uptake and decreasing endogenous glucose production are needed to reduce hyperglycemia…

For the same reason, I think that an insulin arrow between islet and liver is missing in Figure 1.  The liver is an important target of insulin as put in evidence later in the manuscript with the LIRKO mice.

P3 L 90 : a short sentence about the role of miRNAs should be added

P3 L96 : It may be useful to highlight the fact that islets hyperplasia is observed in the LIRKO model (not only high IGF-1 levels).

P3 l101-103 : again, the information about insulin direct effect on liver to decrease glucose production (glycogenolysis and neoglucogenesis) is missing. The indirect effect of a decreased beta-cell mass and function in advanced T2D patients on glucagon hypersecretion plays undoubtedly a role in elevated glucose output, but not alone…

P4  L145 : the comma is misplaced

P7 L297 : The whole paragraph should take place earlier…In addition, the difference between EVs and exosomes is unclear …Are they synonyms?

P7 L 324 : “Figure 2 shows..”  or “the main players are shown…”

P8 L343-344 : a verb is missing

Author Response

Manuscript ID 1531325

REFEREE 2

 Comments and Suggestions for Authors

             Answers by the authors

The review “Intestine-pancreatic beta cell intercommunication: a new approach to understand diabetes” presents an overview of the crosstalk between beta-cells and metabolic and non-metabolic tissues and gives an update of the state of the art about microbiota role in beta-cell biology.

The review is comprehensive and clearly written, although the modifications suggested below could significantly improve the manuscript.

We would like to thank the Reviewer for the careful reading of the manuscript, as well as the encouraging comments and helpful suggestions.

All changes have been highlighted in red. Deleted sentences or paragraphs are indicated in red and crossed out.

  1. The title: it is focused on Intestine-pancreatic intercommunication, but the first 4,5 pages are about adipose tissue, liver and muscle dialogue with beta-cells. This is not a criticism because this part of the review is relevant and interesting, however it questions the relevance of the title. I would suggest something more general such as: “Multi-organ crosstalk with endocrine pancreas: a focus on how gut microbiota shapes pancreatic beta-cells”, for example.

We totally agree with the Reviewer that the title chosen does not accurately reflect the content of the manuscript. For this reason, we have changed the title to a more appropriate one.

  1. Then, the first appearance of the EVs is in Figure 1 legend (abstract excluded), which is inappropriate.

A specific paragraph on Extracellular Vesicles should take place in the Introduction, to define what it is…It seems more relevant to explain it first as it concerns the 4 tissues of interest (fat, muscle liver and gut) than to add information about EVs in page 7, in a specific Gut-centred paragraph. However, the information about microbiota-produced EVs is relevant in the 4.2 paragraph, of course.

We are totally agree with the Reviewer regarding the importance of defining extracellular vesicles in the Introduction in order to make the data shown below more understandable for the reader. Moreover, a table (Table 1) has also been added for the same reason.

We hope this new version will be more accurately and will reflect the information included along the manuscript.

In addition, the fact that some EVs are microbiota-produced is very important and may deserve a specific paragraph with appropriate references.

As suggested by the Reviewer, this issue has been explained in more detail and new references have been included in the revised version of the manuscript (please see P9, L374-425). 

  3. Figure 2: the figure itself and the legend present only negative effects of SCFAs and EVs, which doesn’t correspond to the text. It is unclear if the figure displays a pathologic condition such as obesity…And the blue arrow with ‘prebiotics and probiotics” is confusing, although I understand it is a way to counterbalance the deleterious effects of dysbiosis…

Thanks to the reviewer for her/his helpful comment. In this regard, we have rewritten the figure legend 2 in order to explain more precisely the content of the figure. In addition, we have included the origin of the different extracellular vesicles. In one hand, the microbiome-derived extracellular vesicles or MEVs and the intestinal extracellular vesicles.

Please note that Figure 2 has been renumbered in the revised version of the manuscript as Figure 3 because a new figure has been included.

Besides these general comments about the manuscript organisation, I have some specific points to discuss.

  1. P2, L52-53 : I feel that the sentence doesn’t take into account the specific effect of insulin on the liver, that is to say the decreased hepatic glucose production. Both improving glucose uptake and decreasing endogenous glucose production are needed to reduce hyperglycemia…

This sentence has been rewritten to explain more accurately the specific effect of insulin on liver.

For the same reason, I think that an insulin arrow between islet and liver is missing in Figure 1.  The liver is an important target of insulin as put in evidence later in the manuscript with the LIRKO mice.

Figure 1 has been modified following Referee’s suggestions.

  1. P3 L 90: a short sentence about the role of miRNAs should be added

In agreement with the Reviewer, the function of miRNAs is briefly explained in this paragraph (please see P4, L133-134).

  1. P3 L96: It may be useful to highlight the fact that islets hyperplasia is observed in the LIRKO model (not only high IGF-1 levels).

We have followed Referee’s suggestion (please see P4, L144).

  1. P3 L101-103: again, the information about insulin direct effect on liver to decrease glucose production (glycogenolysis and neoglucogenesis) is missing. The indirect effect of a decreased beta-cell mass and function in advanced T2D patients on glucagon hypersecretion plays undoubtedly a role in elevated glucose output, but not alone…

The sentence has been rewritten to more properly express the net metabolic effect of the coordinated action of counter-regulatory hormones in liver (please, see P4, L148-151).

  1. P4  L145: the comma is misplaced

We have replaced the comma adequately.

  1. P7 L297: The whole paragraph should take place earlier…In addition, the difference between EVs and exosomes is unclear …Are they synonyms?

As indicated in point number 2, the concept of extracellular vesicles has been included in the introduction. The new definition is more precise now and we hope it may be considered acceptable.

Regarding the question whether EVs and exosomes are synonyms, the answer is no, they are not exactly synonyms. Extracellular vesicles or EVs are double-membrane structures which are produced by cells. Within EV denomination it is included exosomes, microvesicles or even apoptotic bodies. Exosomes are EVs of ~50–150 nm in diameter. Microvesicles are similar in structure, content, and function to exosomes, but are larger in diameter (~100–1,000 nm). Apoptotic bodies are formed in the process of cell death from fragments of the parental cell and range a wide size (~100–5,000 nm in diameter). In acknowledgment that samples of EVs described in the research literature as exosomes or microvesicles may include other vesicles of similar size because of the practical difficulties inherent in distinguishing exosomes from other small EVs in a biofluid (Witwer, K.W.; Théry, C. Extracellular vesicles or exosomes? on primacy, precision, and popularity influencing a choice of nomenclature. J Extracell Vesicles 2019, 8:1648167), we decided herein to group in the term EVs all types of small vesicles.

  1. P7 L 324: “Figure 2 shows..”  or “the main players are shown…”

We have modified accordingly

  1. P8 L343-344: a verb is missing

We have modified accordingly to reviewer´s suggestion

Round 2

Reviewer 1 Report

It is clear to see that the quality of manuscript has been markedly improved. Thank you.